# Soluble stroma-related biomarkers of pancreatic cancer

Andrea Resovi[1,†], Maria Rosa Bani[1,†], Luca Porcu[2], Alessia Anastasia[1], Lucia Minoli[3], Paola Allavena[4], Paola Cappello[5,6,7] (iD), Francesco Novelli[5,6,7], Aldo Scarpa[8], Eugenio Morandi[9], Anna Falanga[10], Valter Torri[2], Giulia Taraboletti[1,‡], Dorina Belotti[1,*,‡] (iD) & Raffaella Giavazzi[1,‡,**] (iD)

## Abstract

The clinical management of pancreatic ductal adenocarcinoma (PDAC) is hampered by the lack of reliable biomarkers. This study investigated the value of soluble stroma-related molecules as PDAC biomarkers. In the first exploratory phase, 12 out of 38 molecules were associated with PDAC in a cohort of 25 PDAC patients and 16 healthy subjects. A second confirmatory phase on an independent cohort of 131 PDAC patients, 30 chronic pancreatitis patients, and 131 healthy subjects confirmed the PDAC association for MMP7, CCN2, IGFBP2, TSP2, sICAM1, TIMP1, and PLG. Multivariable logistic regression model identified biomarker panels discriminating respectively PDAC versus healthy subjects (MMP7 + CA19.9, AUC = 0.99, 99% CI = 0.98–1.00) (CCN2 + CA19.9, AUC = 0.96, 99% CI = 0.92–0.99) and PDAC versus chronic pancreatitis (CCN2 + PLG+FN+Col4 + CA19.9, AUC = 0.94, 99% CI = 0.88–0.99). Five molecules were associated with PanIN development in two GEM models of PDAC (PdxCre/LSL-Kras[G12D] and PdxCre/LSL-Kras[G12D/+]/LSL-Trp53[R172H/+]), suggesting their potential for detecting early disease. These markers were also elevated in patient-derived orthotopic PDAC xenografts and associated with response to chemotherapy. The identified stroma-related soluble biomarkers represent potential tools for PDAC diagnosis and for monitoring treatment response of PDAC patients.

**Keywords** circulating biomarkers; early diagnosis; pancreatic cancer; treatment evaluation; tumor microenvironment
**Subject Categories** Biomarkers & Diagnostic Imaging; Cancer; Post-translational Modifications, Proteolysis & Proteomics

## Introduction

Pancreatic ductal adenocarcinoma (PDAC) is one of the most aggressive epithelial malignancies, with a 5-year survival rate of 6% (Tempero *et al*, 2013). Although progression from tumor initiation to advanced invasive cancer may take up to about 10 years (Yachida *et al*, 2010), PDAC is often diagnosed at an advanced stage, because of non-specific symptomatology, the absence of effective imaging tests to identify early disease, and the lack of specific and sensitive diagnostic circulating biomarkers (Korc, 2007).

Late diagnosis of PDAC leads to a limited therapeutic time window during which serological markers capable of monitoring treatment effectiveness could change the fate of PDAC patients.

The biomarker CA19.9, currently used to detect and monitor PDAC, is not sufficiently sensitive and specific to have reliable diagnostic value. In addition, it is not expressed in approximately 20% of the Lewis antigen-negative population. There are also racial and sex variations in CA19.9 expression with the highest levels in Caucasians (Tempero *et al*, 2013). Recent proteomic studies have identified circulating molecules or autoantibodies that are up-regulated in PDAC, but few have been investigated further as a serological diagnostic or prognostic biomarker for the disease (Brand *et al*, 2011; Capello *et al*, 2013, 2017; Chan *et al*, 2014; Shaw *et al*, 2014; Zhang *et al*, 2014; Balasenthil *et al*, 2017).

1 Laboratory of Biology and Treatment of Metastasis, Department of Oncology, IRCCS-Istituto di Ricerche Farmacologiche Mario Negri, Bergamo and Milan, Italy
2 Laboratory of Methodology for Clinical Research, Department of Oncology, IRCCS-Istituto di Ricerche Farmacologiche Mario Negri, Milan, Italy
3 Mouse and Animal Pathology Lab, Fondazione Filarete and Department of Veterinary Pathology, University of Milan, Milan, Italy
4 Department of Immunology and Inflammation, IRCCS-Humanitas Clinical and Research Center, Rozzano, Italy
5 CERMS, AOU Città della Salute e della Scienza, Turin, Italy
6 Department of Molecular Biotechnology and Health Sciences, University of Turin, Turin, Italy
7 Molecular Biotechnology Center, Turin, Italy
8 Department of Pathology and Diagnostic, University and Hospital Trust of Verona, Verona, Italy
9 Chirurgia IV, Presidio Ospedaliero di Rho, ASST Rhodense, Milano, Italy
10 Department of Immunohematology and Transfusion Medicine, Thrombosis and Hemostasis Center, Hospital Papa Giovanni XXIII, Bergamo, Italy
*Corresponding author. Tel: +39 035 42131; Fax: +39 035 319331; E-mail: dorina.belotti@marionegri.it
**Corresponding author. Tel: +39 02 39014732; Fax: +39 02 39014734; E-mail: raffaella.giavazzi@marionegri.it
†These authors contributed equally to this work as first authors
‡These authors contributed equally to this work as senior authors

PDAC is notable for its desmoplastic stromal reaction and prominent extracellular matrix (ECM) deposition (Whatcott et al, 2015). Stromal elements and extracellular matrix remodeling have a role in PDAC progression and, ultimately, chemotherapy delivery and activity (Feig et al, 2012). The microenvironment changes from normal to a tumor-supportive state, favoring tumor growth and invasion. It has been suggested that desmoplasia might have a prognostic role since fibrosis, stromal abundance, and reactivity have been correlated with shorter survival in patients with resected PDAC (Erkan, 2013). The abundant stroma is one of the main reasons for the limited drug response of PDAC (Neesse et al, 2011).

Based on these observations, we hypothesized that the tumor microenvironment might be a source of circulating molecules exploitable as diagnostic biomarkers and as endpoints of target therapies. Since stromal modifications occur early in tumorigenesis and persist in advanced tumors, stroma-related circulating molecules might have great potential for the diagnosis of PDAC identified at a stage in which the disease is still operable.

In this study, we combined multiple approaches to investigate circulating stroma-related molecules as PDAC diagnostic biomarkers and as endpoints for assessing the effectiveness of treatment. Thirty-eight stroma-associated potential circulating biomarkers, including extracellular matrix proteins and proteolytic fragments, matrix-degrading enzymes and their inhibitors, growth factors, antiangiogenic factors, and adhesion molecules, were selected from previous proteomic analyses (Yu et al, 2005; Bloomston et al, 2006; Faca et al, 2008; Kojima et al, 2008; Fiedler et al, 2009; Rong et al, 2010; Xue et al, 2010; Pan et al, 2011) and measured in the plasma of PDAC patients, healthy controls, and chronic pancreatitis patients. Selected candidate molecules were further validated in genetically engineered mouse models of PDAC with mutated Kras (KC mice) or with mutated Kras and TP53 (KPC mice) and associated with PDAC initiation and progression (PanIN-PDAC) (Hingorani et al, 2003, 2005; Capello et al, 2013), as well as in patient-derived PDAC xenografts (PDAC-PDX), where their levels correlated with tumor burden and response to treatment.

## Results

### Selection of candidate PDAC stroma-related biomarkers

Tumor-stroma-associated PDAC biomarkers were selected from eight proteomic studies on circulating proteins that are differentially expressed in PDAC and healthy subjects (Yu et al, 2005; Bloomston et al, 2006; Faca et al, 2008; Kojima et al, 2008; Fiedler et al, 2009; Rong et al, 2010; Xue et al, 2010; Pan et al, 2011). Thirty-eight candidates were selected because they were found in at least two independent analyses and/or were related to tumor/stroma interaction by Gene Ontology. These data were integrated with manually curated additional information from the literature. The selected molecules are listed in Table EV1 and include extracellular matrix proteins and proteolytic fragments, matrix-degrading enzymes and their inhibitors, growth factors, angiogenesis regulatory factors, and adhesion molecules.

## Analysis of circulating PDAC stroma-related biomarkers in patients

### First exploratory phase

The levels of the 38 selected candidate biomarkers were analyzed in the plasma of patients with histologically verified PDAC (n = 25) and in healthy controls (n = 16) (cohort no. 1 in Table 1). The concentrations are shown in Table EV2.

We identified six clusters of biomarkers that are as correlated as possible with each other and as uncorrelated as possible with biomarkers in other clusters (Table 2). The plasma levels of the molecules in these clusters are shown in Fig 1. At univariate logistic regression, three clusters (clusters 3, 4, and 6) were significantly associated with the presence of PDAC (respectively $P = 0.007$, $P = 0.005$, and $P = 0.07$).

Molecules in these clusters (with the exception of Lam-P1 for which commercially available kits had been discontinued) were selected for further analysis.

### Second confirmatory phase

Molecules in clusters 3, 4, and 6 (TIMP1, sICAM1, MMP7, PICP, PLG, TSP2, IGFBP2, FN, PINP, CCN1, CCN2, Col4) were further analyzed in a larger independent cohort of PDAC patients (n = 131), pancreatitis patients (n = 30), and sex-matched healthy individuals (n = 131) (cohort no. 2 in Table 1). The distribution of the twelve molecules in the second cohort of patients is shown in Table EV3.

Seven (TIMP1, sICAM1, MMP7, TSP2, PLG, IGFBP2, and CCN2) of the 12 molecules were significantly up-regulated in PDAC patients compared to healthy controls ($P < 0.001$) (Fig 2A; Tables EV3 and EV4). The differences between PDAC and healthy subjects were excellent for MMP7 (AUC = 0.98) and good for CCN2 (AUC = 0.86), which demonstrated a discriminatory ability similar to CA19.9 (AUC = 0.87), while IGFBP2 and TIMP1 (AUC = 0.82), TSP2 (AUC = 0.78), sICAM1 (AUC = 0.77), and PLG (AUC = 0.66) had a weaker discriminatory ability (Fig EV1 and Table EV4). The seven molecules were confirmed to be significantly up-regulated also at early stages (stages IA, IB, and IIA) when tumor is confined to the pancreas and is not spread to nearby lymph nodes (N0) or to distant

Table 1.  Clinical characteristics of the study population.

| | Number of cases | Gender (%) | Age median (Range) | PDAC stage (%) |
|---|---|---|---|---|
| Cohort 1 (Exploratory phase) | | | | |
| Healthy | n = 16 | M (62.5) F (37.5) | 59 (54–65) | Stage IA (16) Stage IIA (24) Stage IIB (56) ND (4) |
| PDAC | n = 25 | M (40) F (60) | 50 (47–82) | |
| Cohort 2 (Confirmatory phase) | | | | |
| Healthy | n = 131 | M (49.6) F (50.4) | 55 (44–66) | Stage IA (1.5) Stage IB (0.8) Stage IIA (16.8) Stage IIB (66.4) Stage III (0.8) Stage IV (2.3) ND (11.4) |
| Chronic pancreatitis | n = 30 | M (73.3) F (26.7) | 53 (34–79) | |
| PDAC | n = 131 | M (49.6) F (50.4) | 70 (38–88) | |

ND, not determined; M, males; F, females.

**Table 2.  Clusters of biomarkers associated with PDAC.**

| Cluster No. | Molecules | $R_o^{2a}$ | $R_n^{2b}$ | $(1 - R_o^2)/(1 - R_n^2)$ ratio | $P$ value[c] |
|---|---|---|---|---|---|
| 1 | MMP12 | 0.810 | 0.087 | 0.208 | 0.364 |
|   | MMP13 | 0.790 | 0.117 | 0.238 | |
|   | IGFBP4 | 0.864 | 0.316 | 0.199 | |
|   | IGFBP5 | 0.726 | 0.266 | 0.373 | |
|   | SPARC | 0.656 | 0.206 | 0.433 | |
| 2 | ES | 0.581 | 0.086 | 0.459 | 0.176 |
|   | PDGF-BB | 0.776 | 0.253 | 0.299 | |
|   | FGF-2 | 0.848 | 0.087 | 0.166 | |
|   | VEGFA | 0.948 | 0.098 | 0.057 | |
| 3 | TIMP1 | 0.836 | 0.181 | 0.200 | 0.007 |
|   | sICAM1 | 0.535 | 0.210 | 0.588 | |
|   | MMP7 | 0.648 | 0.056 | 0.373 | |
|   | PICP | 0.244 | 0.067 | 0.810 | |
|   | PLG | 0.249 | 0.126 | 0.859 | |
|   | TSP2 | 0.772 | 0.333 | 0.342 | |
| 4 | IGFBP2 | 0.573 | 0.232 | 0.556 | 0.005 |
|   | FN | 0.231 | 0.053 | 0.812 | |
|   | PINP | 0.121 | 0.024 | 0.900 | |
|   | CCN1 | 0.587 | 0.294 | 0.585 | |
|   | CCN2 | 0.627 | 0.097 | 0.413 | |
| 5 | sVCAM1 | 0.606 | 0.042 | 0.412 | 0.332 |
|   | NGAL | 0.606 | 0.024 | 0.404 | |
| 6 | Col4 | 0.715 | 0.085 | 0.311 | 0.070 |
|   | Lam-P1 | 0.715 | 0.063 | 0.304 | |

[a]Squared correlation coefficient between a given biomarker and its own cluster.
[b]The next highest squared correlation coefficient between a given biomarker and any other cluster.
[c]$P$ value from 1 df Wald $\chi^2$ for association with outcome.

sites (M0). AUCs used to estimate the predictive accuracy of distributional models did not change significantly between stages (Fig EV1 and Appendix Table S2). Age difference between PDAC patients and healthy individuals did not affect the significant association between each selected biomarker and PDAC, as assessed by a multivariable logistic regression model, adjusted for age effect (data not shown). No correlation was found between these molecules and CA19.9 (Appendix Fig S1).

Cohort no. 2 also included 30 patients with chronic pancreatitis (CP). The levels of TIMP1, sICAM1, MMP7, IGFBP2, CCN2, and TSP2, though lower than in PDAC, were increased in plasma from CP compared with healthy controls. Significantly lower levels of Col4 and FN were specifically associated with CP ($P < 0.001$). PLG in CP was significantly lower than in PDAC ($P < 0.001$), but not significantly different from healthy subjects ($P < 0.05$) (Fig 2A and Table EV4).

Using a multivariable logistic regression model, we then developed biomarker panels that improved the discovery power of CA19.9 in PDAC versus healthy subjects and PDAC versus CP. In

accordance with the statistical procedure adopted, the resulting panel consisting of MMP7 and CA19.9 statistically better discriminated PDAC versus healthy subjects with an AUC of 0.99 (99% CI = 0.98–1.00) compared with CA19.9 (AUC = 0.87, 99% CI = 0.81–0.93). Similarly, the panel consisting of CCN2 and CA19.9 with an AUC of 0.96 (99% CI = 0.92–0.96) discriminated PDAC versus healthy subjects better than CA19.9 (Fig 2B). Multivariable models defined in the overall population were evaluated by stage. As reported in Appendix Table S2, they confirmed their optimal predictive accuracy without significant interaction between stages.

A panel consisting of CCN2, PLG, FN, Col4, and CA19.9 improved the performance of CA19.9 in distinguishing PDAC from CP. These five biomarkers, which individually (Table EV4) had an AUC of 0.56 (99% CI = 0.41–0.71, CCN2), 0.74 (99% CI = 0.61–0.86, PLG), 0.80 (99% CI = 0.65–0.94, FN), 0.74 (99% CI = 0.59–0.89, Col4), and 0.83 (99% CI = 0.75–0.92, CA19.9), when analyzed in combination showed an AUC of 0.94 (99% CI = 0.88–0.99) indicating a significantly higher capability to discriminate PDAC from CP (Fig 2B). Similar results were obtained separating PDAC by stages (Appendix Table S2).

### Circulating stroma-related molecules in mouse models of PDAC

#### High TIMP1, MMP7, TSP2, CCN2, and ICAM1 in Kras[G12D]- and p53[R172H]-driven PanIN development

We used KC mice expressing the mutation of Kras (Kras[G12D]) in pancreatic progenitor cells and progressing from a healthy condition to different grades of PanIN (PanIN-1A–1B–2–3) (Hingorani *et al*, 2003; Cappello *et al*, 2013) and KPC mice, carrying Kras[G12D/+] and p53[R172H/+] mutations and developing PanINs that ultimately progresses to overt carcinoma (Hingorani *et al*, 2005) to study the importance of the selected biomarkers during PDAC induction.

Plasma was collected at 60, 120, 180, 240, and 330 days of age from KC mice and at 30, 90, and 150 days of age from KPC mice. We measured those biomarkers for which reliable ELISA was commercially available.

Plasma TIMP1, MMP7, and TSP2 levels rose significantly over time in relation to PanIN development in both KC and KPC mice (Fig 3A). At death of KC mice on day 330, when histological analysis of the pancreas confirmed the presence of PanIN-1A, PanIN-1B, and PanIN-2 in, respectively, 2, 2, and 4 lesions (Appendix Fig S2A), MMP7 and TSP2 plasma levels were significantly higher than control PdxCre mice (Fig 3B; KC).

TIMP1, MMP7, and TSP2 levels were also significantly higher in KPC than PdxCre mice at 150 days of age (Fig 3B; KPC).

None of the three molecules were elevated in the plasma of mice with caerulein-induced chronic pancreatitis after 7 weeks of treatments confirming the specific association of high levels of TIMP1, MMP7, and TSP2 with neoplastic transformation (Fig 3B and Appendix Fig S3).

Pancreatic RNA expression analysis was performed for CCN2, IGFBP2, ICAM1, and PLG for which ELISA kits to measure mouse proteins were not available, and for TIMP1, MMP7, and TSP2. Real-time PCR with mouse-specific probes showed that TIMP1, MMP7, TSP2, CCN2, and ICAM1 ($P < 0.05$) were more expressed

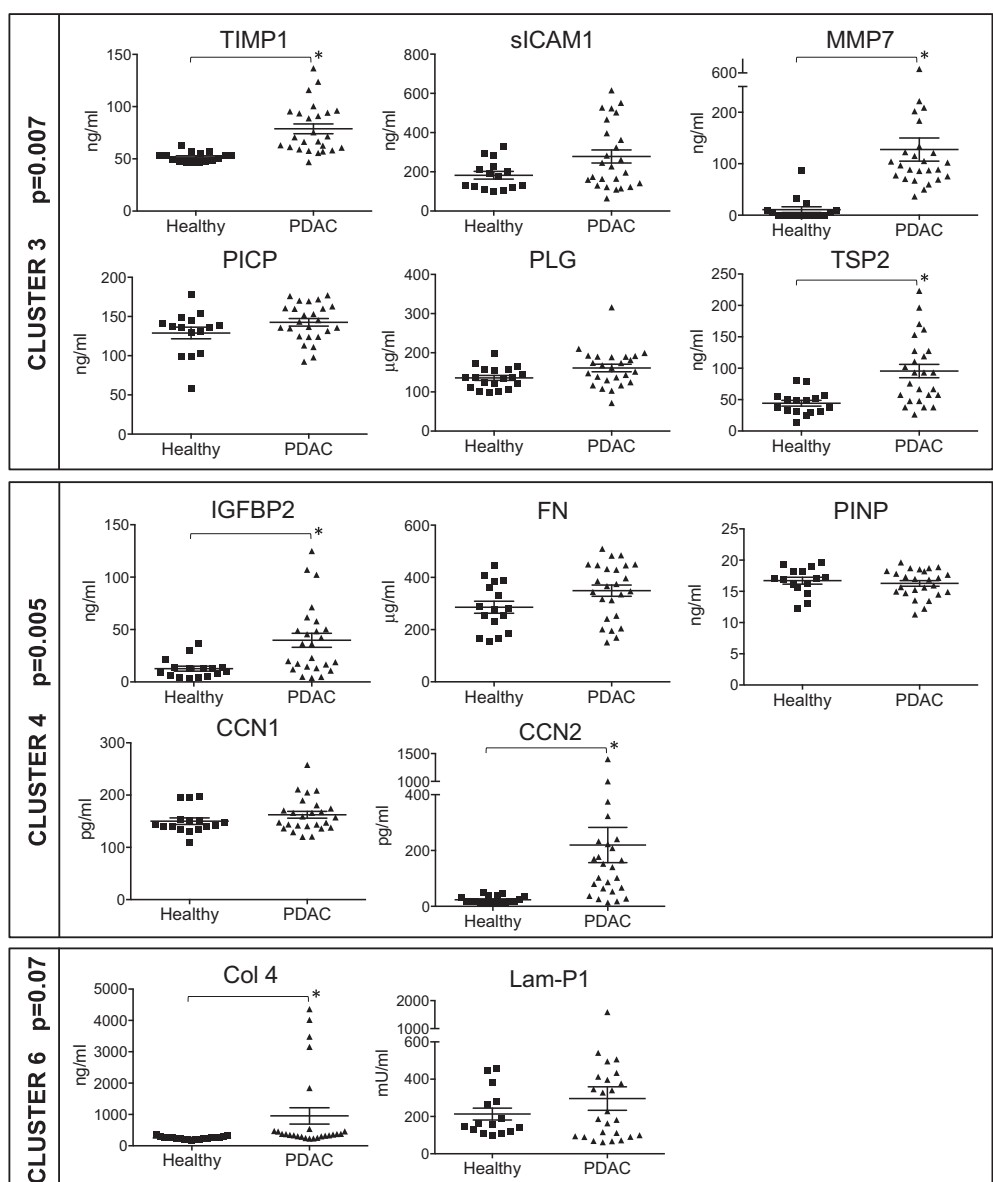

**Figure 1.  Phase I exploratory phase (cohort no. 1): plasma levels of molecules belonging to PDAC-associated clusters.**

Plasma levels of the 13 biomarkers belonging to clusters 3, 4, and 6 (see Table 2) in healthy subjects ($n = 16$) and PDAC patients ($n = 25$), *$P < 0.01$ (Mann–Whitney). Data are expressed as a scatter plot, mean $\pm$ SEM. $P$-values (for each cluster) were calculated with the Wilcoxon rank-sum test and indicate a significant association between each cluster and PDAC.

in KC than in control PdxCre mice at 330 days of age (Appendix Fig S2B). TIMP1, MMP7, TSP2, and CCN2 expression in PanIN lesions was confirmed by immunohistochemistry. TIMP1, MMP7, TSP2, and CCN2 staining was typically low in PdxCre pancreas, while a more intense staining was observed in PanIN lesions (Fig 3C).

### TIMP1, MMP7, TSP2, CCN2, and ICAM1 are elevated in mice with patient-derived PDAC xenografts

We measured TIMP1, MMP7, and TSP2 using mouse-specific ELISA in three PDAC-PDX (HuPa4, HuPa8, and HuPa11) growing

orthotopically in the pancreas of SCID mice. These tumors are characterized by relevant amounts of host murine stroma (Appendix Fig S4) supporting their use for stroma-derived biomarker validation. In all the PDAC-PDX models, circulating mouse TIMP1, MMP7, and TSP2 levels were significantly higher than in healthy mice ($P < 0.05$) (Fig 4A) paralleling the tumor growth in the pancreas as shown by the significant correlation with tumor burden measured by MRI (TIMP1, $r = 0.68$; MMP7, $r = 0.60$ and TSP2, $r = 0.82$) (Fig 4B).

TIMP1, TSP2, and CCN2 but not MMP7 mRNA was highly expressed in HuPa4, HuPa8, and HuPa11 (PDAC-PDX versus healthy

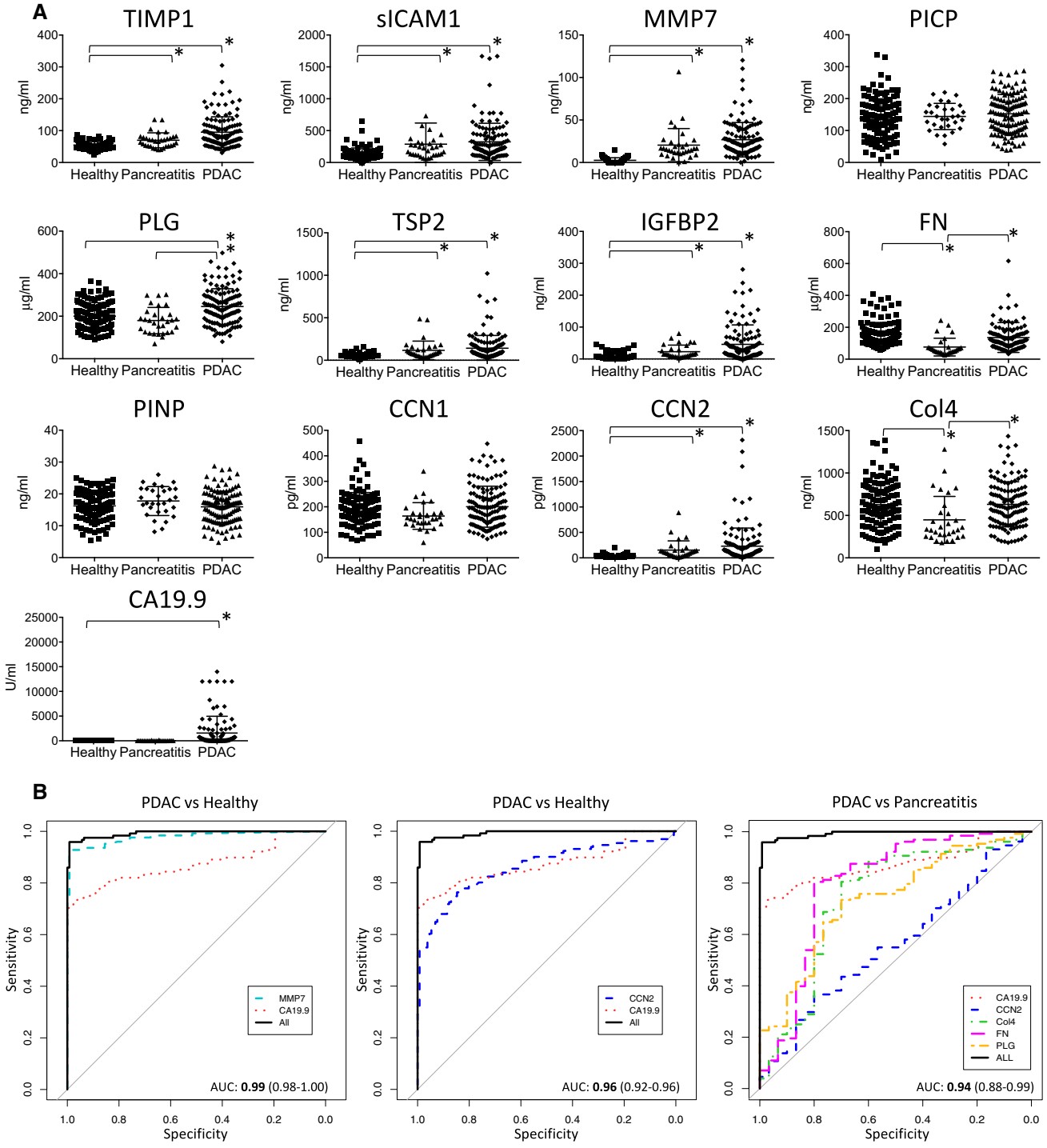

**Figure 2. Phase II confirmatory phase (cohort no. 2): plasma levels and AUC values of selected molecules.**

A   Plasma levels of selected candidate biomarkers analyzed in healthy subjects (*n* = 131), pancreatitis patients (*n* = 30), and PDAC patients (*n* = 131). Data are expressed as a scatter plot, mean ± SEM, \**P* < 0.001 (Wilcoxon rank-sum test).

B   Receiver operator characteristic (ROC) curves of the single biomarkers and of biomarker panels (indicated with All) for diagnosis of PDAC versus healthy controls and PDAC versus pancreatitis. Areas under the curve (AUC) with 99% CI are presented.

mice *P* < 0.05) (Fig 4C). In agreement, immunohistochemical analysis highlighted a strong expression of TIMP1, TSP2, and CCN2 proteins in the tumor stroma (Fig 4D). ICAM1 was more expressed in

all the three PDAC-PDX compared to healthy mice (*P* < 0.05) while IGFBP2 mRNA levels were significantly higher only in HuPa4 (*P* < 0.05), and PLG was not increased in PDAC-PDX (Fig 4C).

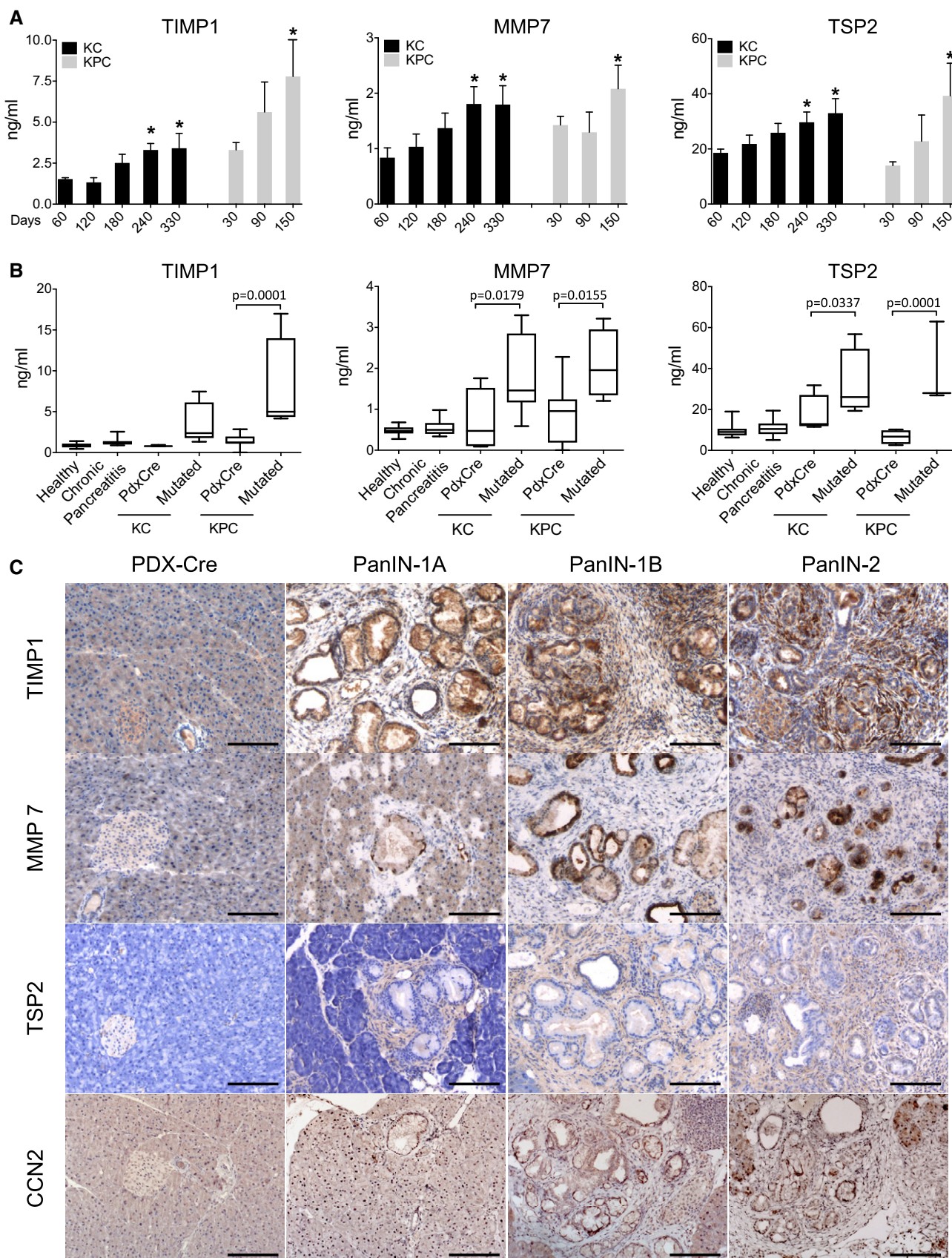

**Figure 3.**

**Figure 3.   Biomarkers in PdxCre/LSL-Kras[G12D] (KC) and Kras[G12D]/Trp53[R172H] (KPC) GEM models.**

A   Levels of TIMP1, MMP7, and TSP2 in plasma of KC mice ($n$ = 7–9) at 60, 120, 180, 240, and 330 days of age and of KPC mice ($n$ = 3–8) at 30, 90, and 150 days of age (mean ± SEM). *$P$ < 0.05 (Mann–Whitney). The exact $n$ and $P$-values are indicated in Appendix Table S3A.

B   Levels of TIMP1, MMP7, and TSP2 in plasma of healthy mice ($n$ = 15), mice with chronic pancreatitis at 150 days of age ($n$ = 19), KC mice at 330 days of age (control PdxCre $n$ = 3–4; KC $n$ = 7–8), and KPC mice at 150 days of age (control PdxCre $n$ = 4–7; KPC $n$ = 4). The exact $n$ is indicated in Appendix Table S3B. Box plots extend from 25th to 75th percentiles, whiskers extend from min to max, and horizontal lines indicate median. $P$-values were calculated with one-way ANOVA with Tukey's multiple comparison test.

C   Histological analysis of pancreas from PdxCre and KC mice with different grades of PanIN lesions at 330 days of age. Anti-TIMP1, anti-MMP7, anti-TSP2, and anti-CCN2 staining of a representative KC PanIN lesion (200×, scale bars: 100 μm).

### TIMP1, MMP7, and TSP2 as biomarkers of treatment response in PDAC-PDX models

TIMP1, MMP7, and TSP2 increased over time in plasma of mice bearing HuPa8 (Fig 5A). To analyze the relevance of the selected stroma-related molecules as markers of drug response, we treated HuPa8-bearing mice with gemcitabine monotherapy or combined with albumin-bound paclitaxel (NAB-P) to reproduce clinical studies (Von Hoff *et al*, 2013; Goldstein *et al*, 2015).

Gemcitabine as single agent had no worthwhile effect on HuPa8 growing in the mouse pancreas, but the combination with NAB-P significantly inhibited tumor growth (Fig 5B and C).

TIMP1, MMP7, and TSP2 were measured before, during, and after treatments (Fig 5D). Their plasma levels reflected tumor response to treatment, matching the tumor burden changes measured by MRI over time. Differences in the biomarker plasmatic levels became evident after four treatments, paralleling treatment response. After eight treatments, TIMP1, MMP7, and TSP2 were significantly lower ($P$ < 0.05) in the responding (gemcitabine + NAB-P group) compared to the non-responding tumors (gemcitabine group), thus suggesting they were potential markers for monitoring tumor response and treatment efficacy.

## Discussion

Tumor progression relies on the creation of a favorable microenvironment involving not only cancer cells, but also stroma and immune/inflammatory cells. This is particularly true for PDAC (Erkan, 2013). The critical roles of stroma and extracellular matrix remodeling in determining PDAC initiation, growth, and metastasis and in hampering drug response have suggested possibilities for using stroma-related molecules as diagnostic and prognostic biomarkers. This study identified seven stroma-related biomarkers for PDAC and three biomarker panels able to distinguish PDAC patients from healthy individuals or from pancreatitis patients.

In an initial search, we selected 38 stroma-related molecules from proteomic studies because of their strong relevance in tumor/

stroma interaction. The individual relationship of each of these molecules with PDAC has previously been shown (Lekstan *et al*, 2012; Park *et al*, 2012; Nixon *et al*, 2013; Poruk *et al*, 2014; Jenkinson *et al*, 2015); however, our study is the first one to measure and compare all these molecules together in order to select the best candidate biomarkers. From this large number of candidates, TIMP1, sICAM1, MMP7, IGFBP2, CCN2, TSP2, and PLG emerged as PDAC biomarkers, being associated with the presence of PDAC in two independent cohorts of patients and two autonomous sequential analyses.

The fact that in the second confirmatory phase the seven selected molecules were significantly higher in PDAC patients than in healthy individuals validates the statistical approach based on the clustering method used in the first exploratory phase of the study. Here, the principal components-based clustering method used (Black & Watanabe, 2011) was chosen as a data reduction method to control type I and type II error rates better than univariate analysis of individual markers, when analyzing multiple molecules in a small cohort of patients.

Multivariable logistic regression model identified MMP7 plus CA19.9 and CCN2 plus CA19.9 as the circulating biomarker panels that better discriminate PDAC patients from healthy subjects. MMP7, also known as matrilysin, is a well-established inducer of acinar-to-ductal metaplasia (Crawford *et al*, 2002; Sawey *et al*, 2007), and it was found focally up-regulated in PanIN lesions of the Kras-driven GEM model of PDAC (Hingorani *et al*, 2003). Although MMP7 involvement in PDAC initiation and progression has been reported (Fukuda *et al*, 2011), our results showing high MMP7 plasma levels in association with early tumor progression in PDAC-GEM models and in patients with PDAC at stages IA, IB, and IIA confirm its value as a potential early diagnostic biomarker. CCN2, also known as connective tissue growth factor (CTGF), is overexpressed by activated pancreatic stellate cells in the earliest stages of PDAC. It promotes local desmoplasia, tumor survival, and metastasis (Leask, 2009). Soluble CCN2 in plasma, serum, and urine has promising applicability as biomarker in different types of chronic diseases with fibrosis-related complications (Dendooven *et al*, 2011). While CCN2 has been shown up-regulated in PDAC tumors,

**Figure 4.   Biomarkers in PDAC-PDX growing orthotopically in mouse pancreas.**

A   Levels of murine TIMP1, MMP7, and TSP2 in plasma of mice bearing PDAC-PDX (HuPa4, HuPa8, and HuPa11) growing orthotopically in the pancreas (mean ± SEM; $n \geq 3$ for each group), *$P$ < 0.05 (Mann–Whitney). The exact $n$ and $P$-values are indicated in Appendix Table S4A.

B   Correlations between the levels of the three selected biomarkers and the tumor volume in mice bearing HuPa8. Pearson coefficient (r).

C   Expression of murine TIMP1, MMP7, TSP2, CCN2, ICAM1, IGFBP2, and PLG analyzed in tumors from pancreas (HuPa4, HuPa8, and HuPa11) by RT–PCR. The expression level of target genes was normalized to the geometric median of β-actin and GAPDH housekeeping genes and expressed as $2^{-\Delta\Delta CT}$ (mean ± SEM, *$P$ < 0.05; Healthy $n$ = 7; HuPa4, HuPa8, and HuPa11 $n$ = 4) (Mann–Whitney). The exact $P$-values are indicated in Appendix Table S4B.

D   Histological analysis of representative PDAC-PDX (HuPa4, HuPa8, and HuPa11). Hematoxylin–eosin, anti-TIMP1, anti-MMP7, anti-TSP2, and anti-CCN2 staining of PDAC-PDX tumors (200×, scale bars: 100 μm).

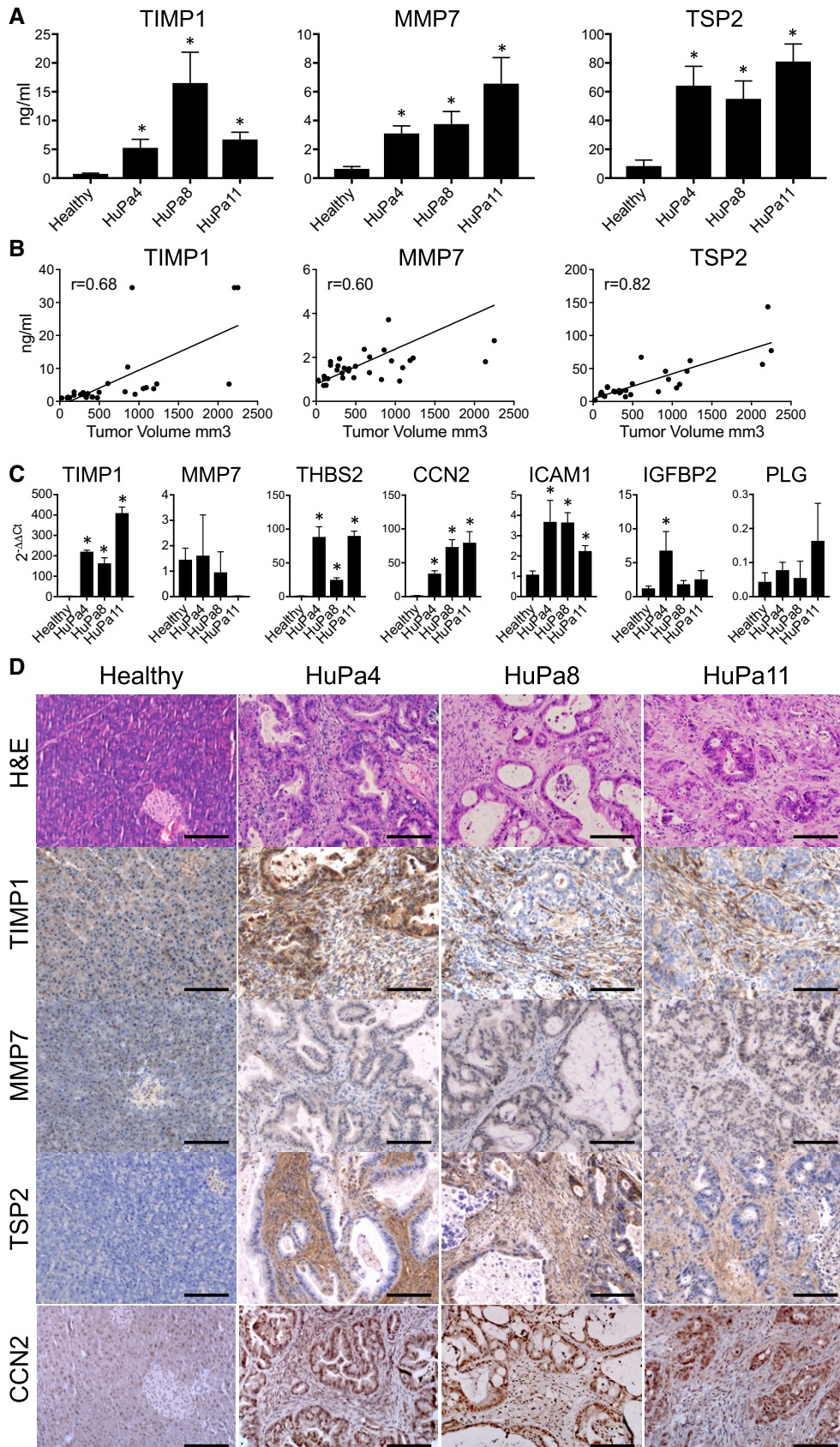

**Figure 4.**

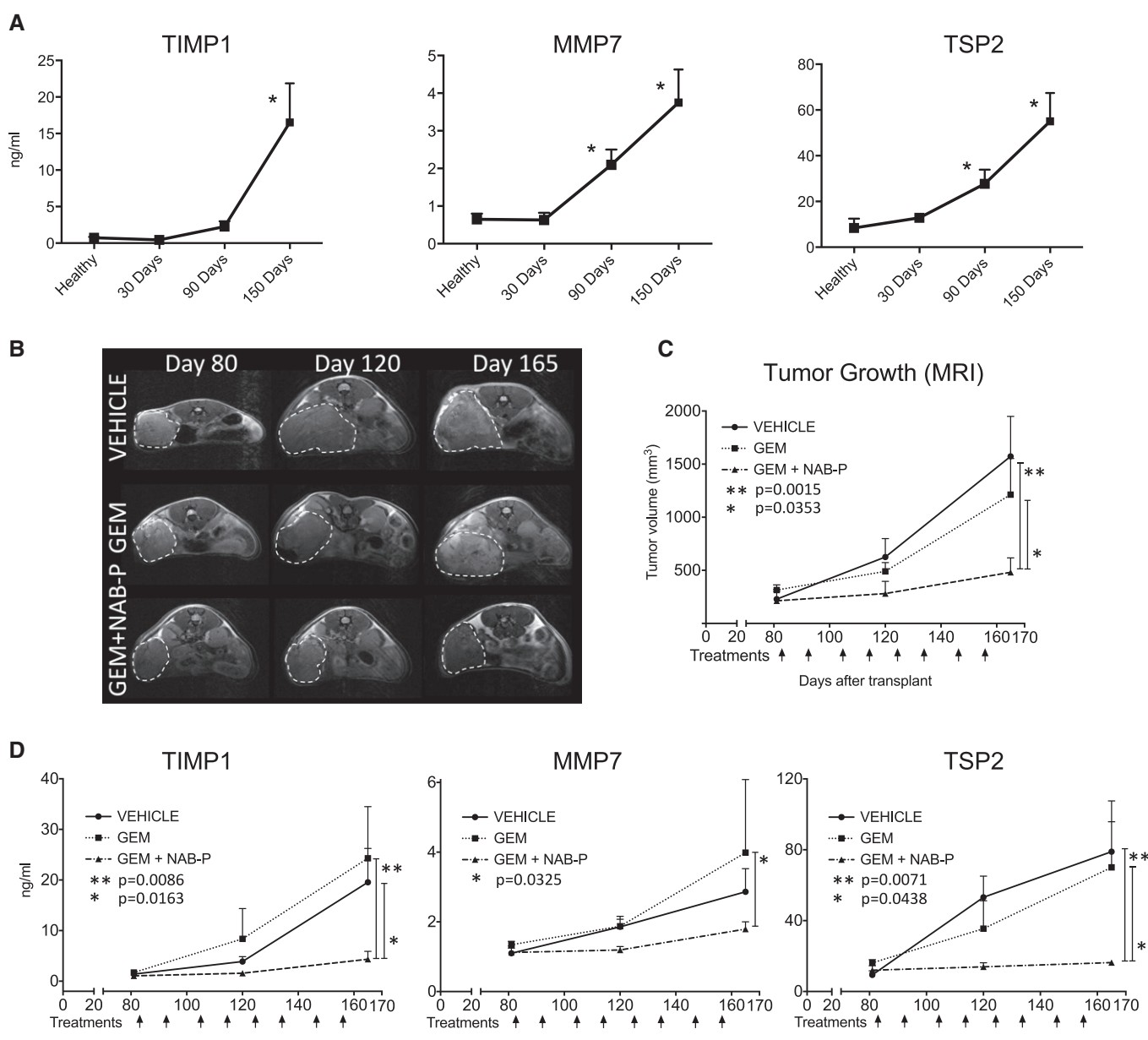

**Figure 5. Murine circulating TIMP1, MMP7, and TSP2 as biomarkers of treatment response in PDAC-PDX models.**

Responsiveness to gemcitabine (GEM), either alone or combined with Nab–paclitaxel (NAB-P), was evaluated in mice bearing orthotopic HuPa8 in the pancreas. Nab–paclitaxel (25 mg/kg) and gemcitabine (150 mg/kg) were given i.v. on days 1 and 8 of each 21-day cycle (for a total of 8 treatments).

A   Levels of TIMP1, MMP7, and TSP2 in plasma of mice bearing orthotopic HuPa8 at 30, 90, and 150 days after tumor transplantation (mean ± SEM; $n \geq 4$ for each group). *$P < 0.05$ (Mann–Whitney). The exact $n$ and $P$-values are indicated in Appendix Table S5A.

B   Magnetic resonance imaging of HuPa8. Representative images are shown; the white dotted lines indicate tumor masses.

C   Tumor growth over time measured by MRI; each black arrow indicates one treatment (mean ± SEM; $n = 4$ for each group; two-way ANOVA with Tukey's multiple comparison test).

D   Levels of the three selected biomarkers in plasma of HuPa8-bearing mice, prior to (day 80; at randomization), during (day 120), and after (day 165) treatments (mean ± SEM; $n \geq 4$ for each group; two-way ANOVA with Tukey's multiple comparison test). The exact $n$ is indicated in Appendix Table S5B.

our study for the first time demonstrates its utility as a circulating PDAC biomarker in patients (Bennewith *et al*, 2009; Neesse *et al*, 2011).

Both MMP7 and CCN2 have been associated with poor prognosis in different types of cancer (Klupp *et al*, 2016; Sun *et al*, 2017)

and with fibrotic or inflammatory diseases (Dendooven *et al*, 2011; Irvine *et al*, 2016). However, our results suggest that an improvement in the specificity of MMP7 and CCN2 results from the inclusion of additional biomarkers. In particular, it needs to be emphasized that the proposed panels of molecules confirmed

greater potential as diagnostic biomarkers for PDAC than any single molecule. MMP7 and CCN2 complemented well the performance of CA19.9 in distinguishing PDAC from healthy subjects. Further addition of PLG, FN, and Col4 to the CCN2 and CA19.9 panel increased the performance of CA19.9 in distinguishing PDAC from CP.

The identification of early diagnostic biomarkers for PDAC is hampered by the fact that most of the clinical samples come from patients with advanced high-grade lesions. The use of *in vivo* preclinical models that faithfully reproduce the complexity of the stroma in PDAC early development is therefore fundamental for biomarker discovery and drug testing.

GEM models of PDAC are unique tools to analyze proteins associated with pre-neoplastic and early-phase tumors, integrating data obtained from patients (Ligat *et al*, 2015). Our findings that tissue TIMP1, MMP7, TSP2, and CCN2 and plasma TIMP1, MMP7, and TSP2 were elevated in association with PanIN development, in both KC and KPC mice, suggest that they appear in the early pre-invasive stage of the disease and can truly detect PDAC prior to the development of clinical symptoms and in lesions with different mutational status.

The value of TIMP1, MMP7, and TSP2 as circulating biomarkers was corroborated in PDAC-PDX-bearing mice where the plasma levels of these molecules reflected the high levels found in the corresponding patient's plasma (data not shown) and correlated with tumor burden over time. Immunostaining with specific anti-mouse antibodies and probes allowed to clarify the origin of these biomarkers and revealed that tissue TIMP1 and TSP2 paralleled their plasma level. Notably, no increase in MMP7 expression was detected by RT–PCR, suggesting that the increased levels of MMP7 detected in the plasma of tumor-bearing mice might reflect a systemic response of the host to the tumor, in line with the known association of MMP7 with inflammation (Fukuda *et al*, 2011).

This also indicates that the host stroma, in both GEM and PDAC-PDX models, reproduces the complexity of the human PDAC microenvironment, undergoing a profound re-organization during tumor progression. Our data in mouse models also indicate that stroma-derived biomarkers can be useful not only for early detection, as evidenced in GEM models, but also for monitoring tumor growth, as evidenced in PDAC-PDX where their levels correlated with tumor burden.

Chronic pancreatitis (CP) is a risk factor for PDAC in humans (Lowenfels *et al*, 1993) and favors the development of PanIN and PDAC in Kras-driven mouse models (Guerra *et al*, 2007). The presence of inflammatory cytokines and high fibrosis create a permissive microenvironment for PanIN. These common pathological features justify the high levels of many of our stroma-related molecules found in the plasma of patients with CP. For example, MMP7 is activated via the Stat3 pathway in CP and involved in PDAC initiation and progression in GEM models of PDAC (Fukuda *et al*, 2011).

Nonetheless, we found three markers, FN, Col4, and PLG, significantly different in PDAC versus pancreatitis. The combination of these three markers with CCN2 and CA19.9 had a significant capability to discriminate between PDAC and CP. Notably, herein the identified CCN2 + PLG+FN+Col4 + CA19.9 panel appeared to have a higher capability in discriminating PDAC from pancreatitis

(AUC = 0.94) than the recently proposed TIMP1 + LRG1 + CA19.9 panel (AUC = 0.89) (Capello *et al*, 2017).

Differently from patients, plasma levels of TIMP1, MMP7, and TSP2 were not increased in the mouse model of caerulein-induced CP and were significantly lower than PanIN. This might be explained by the different pathogenic mechanisms causing CP in patients and mouse model: in patients, usually it takes several years for permanent changes and CP symptoms to occur, for example, after a long-standing overuse of alcohol, while in mice CP is pharmacologically induced and complete damage occurs after few weeks of caerulein treatment.

Patient-derived tumor xenografts that reproduce the morphological and molecular characteristics of the original tumor are being used for drug development (Tentler *et al*, 2012; Ricci *et al*, 2014; Gao *et al*, 2015). Here, we show that TIMP1, MMP7, and TSP2 plasma levels correlated with the response to a Nab–paclitaxel–gemcitabine combined treatment in the PDAC-PDX HuPa8. Stroma-disrupting effects of Nab–paclitaxel in pancreatic cancer have been proposed as marker of drug activity (Alvarez *et al*, 2013). The value of stroma-related molecules as biomarker of response to drugs, particularly those expected to modify the tumor microenvironment (Neesse *et al*, 2013; Masso-Valles *et al*, 2015; Hingorani *et al*, 2016), needs to be further studied.

In conclusion, a PDAC signature of seven circulating stroma-related molecules has been identified and confirmed in two independent cohorts of patients, supporting our hypothesis that stroma remodeling leads to the release of circulating biomarkers. The association of at least five of them with PanIN in mouse models confirmed their potential to detect early lesions. A prospective confirmatory clinical study is required to characterize them as diagnostic PDAC biomarkers complementing CA19.9. Their correlation with tumor burden and drug response in PDAC-PDX models suggests they may also be useful as biomarkers for treatment efficacy in patients.

# Materials and Methods

### Patients

Plasma samples were isolated from peripheral venous blood collected from two independent cohorts of patients with PDAC diagnosed at the Policlinico G.B. Rossi Borgo Roma, Verona (first exploratory phase), and at the Humanitas Research Hospital, Rozzano, Milano (second confirmatory phase). Patients' main characteristics are listed in Table 1. The collection and use of blood samples was approved by the local scientific ethics committees, and patients gave written consent. Criteria for inclusion in the study were as follows: histological diagnosis of PDAC, no previous chemotherapy, and no history of other invasive cancer. Control groups consisted of plasma samples from patients with chronic pancreatitis (CP) and healthy volunteers with no concomitant illnesses. Plasma samples were aliquoted and stored at −80°C until processing.

### Mice and models

Six-week-old female and male PdxCre/LSL-Kras$^{G12D}$ (KC) mice (Hingorani *et al*, 2003), PdxCre/LSL-Kras$^{G12D/+}$/LSL-Trp53$^{R172H/+}$

(KPC) mice (Hingorani *et al*, 2005; Capello *et al*, 2013), and PdxCre control mice were maintained at Biogem (Ariano Irpino, Avellino, IT). Chronic pancreatitis was induced in 10-week-old sex-matched C57BL/6 mice (Envigo, Correzzana, Italy) as previously described (Neuschwander-Tetri *et al*, 2000). Briefly, mice were injected intra-peritoneally with 50 µg/kg of caerulein (AnaSpec, Fremont, CA, USA) 6 times over 5 consecutive hours, three times a week for 7 weeks. The severity of acute injury was initially verified by measuring plasma pancreatic α-amylase using Reflotron tests (Roche, Mannheim, Germany). Pancreatic tissue atrophy and fibrosis were detected by Sirius Red staining at sacrifice.

Three patient-derived PDAC xenografts (HuPa4, HuPa8, and HuPa11) were engrafted in 6- to 8-week-old male severe combined immunodeficiency (SCID) mice (Envigo, Correzzana, Italy) by orthotopic intrapancreas transplantation (manuscript in prepara-tion). The PDAC-PDX models were molecularly and biologically characterized and found to be similar to the original tumor patient. PDAC-PDX were used within the third passage after their first engraftment.

PDAC-PDX growth in the pancreas was monitored by abdominal palpation and by magnetic resonance imaging (MRI) using 7-Tesla BioSpec AVIII system (Bruker Biospin). T2-weighted high-resolution sequences were analyzed using ImageJ software to calculate tumor volume (Cesca *et al*, 2016).

Nab–paclitaxel (Abraxane®, Celgene; 25 mg/kg i.v.) and gemc-itabine (Gemcitabina Teva, Teva, 150 mg/kg i.v.) (HDE: 75 and 450 mg/m$^2$, respectively) were given as single treatment or in combination, on days 1 and 8 of each 21-day cycle for a total of 8 treatments.

Mice were maintained under specific pathogen-free conditions, housed in isolated vented cages, and handled using aseptic proce-dures. Procedures involving animals and their care were conducted in conformity with institutional guidelines that comply with national (Lgs 26/2014) and EU directives laws and policies (EEC Council Directive 2010/63), in line with guidelines for the welfare and use of animals in cancer research (Workman *et al*, 2010). Animal studies were approved by the Mario Negri Institute Animal Care and Use Committee and by Italian Ministry of Health (DM 85/2013-B and Authorization no. 601/2016-PR).

### Biomarker analysis in blood samples

EDTA platelet-poor plasma samples were processed within 30 min of blood drawing, aliquoted, and stored at −80°C until further use. Circulating levels of tumor markers were measured using Luminex-based assays and commercially available ELISA (Millipore S.p.A., Vimodrone, Italy; R&D Systems, Inc., Minneapolis, MN, USA; Cusabio, Wuhan, P.R. China) (Appendix Table S1). Tests were run according to the manufacturer's instructions. Each sample was analyzed in duplicate. Species specificity of the assays for human and murine analytes was verified by the absence of cross-reactivity for, respectively, murine and human plasma.

### Histological and immunohistochemical analysis

PanIN lesions from GEM and tumor specimens from PDAC-PDX were collected, fixed in 10% phosphate-buffered formalin, and embedded in paraffin. Samples were cut into 4-µm-thick sections,

and serial sections from each sample were histochemically stained with H&E, Alcian Pas, and Sirius Red. Samples from GEM were histologically classified at the appropriate stage of tumor progres-sion according to Hingorani *et al* (2003) and graded by the highest-grade component of the lesions. For immunohistochemical analysis, anti-vimentin EPR3776 1:500 (ab92547; Abcam, marker of mesenchymal cells of both human and murine origin), anti-vimentin Sp20 1:500 (RM 92120; Thermo Scientific, marker of mesenchymal cells of human origin), anti-human HLA-A 1:500 (ab52922; Abcam, marker of cells of human origin), anti-TIMP1 1:300 (AF980; R&D), anti-MMP7 1:50 (3801S; Cell Signaling), anti-TSP2 1:25 (PA5-50843; Thermo Scientific), and anti-CCN2 1:100 (ab6992; Abcam) were used.

### Real-time PCR

Total tumor RNA was extracted with TRIzol (Invitrogen), purified, and then reverse-transcribed with the High-Capacity cDNA Archive kit (Applied Biosystems, Monza, Italy), according to the manufac-turer's instructions. TIMP1, MMP7, TSP2, CCN2, ICAM1, IGFBP2, and PLG gene expression was analyzed by real-time qRT–PCR using PrimeTime® Gene Expression Master Mix and murine-specific PrimeTime Std qPCR Assay (Mm.PT.58.30682575 (TIMP1); Mm.PT.58.8800692 (MMP7); Mm.PT.58.31508589 (TSP2)) (IDT, Castenaso, Italy Biosystems) and TaqMan Gene Expression Assay (Mm00516023_m1 (CCN2); Mm00447087_m1 (ICAM1); Mm0049 2632_m1 (IGFBP2); Mm01192931_g1 (PLG)) (Applied Biosystems), normalized against the geometric median of GAPDH (Mm9999 9915_g1) and β-actin (Mm.PT.53a.3177) and expressed as $2^{-\Delta\Delta CT}$.

### Study design and statistical analysis

This study adheres to the guidelines to the Reporting Recommenda-tions for Tumor Marker Prognostic Studies (REMARK; http://www.equator-network.org/reporting-guidelines/reporting-recommenda tions-for-tumour-marker-prognostic-studies-remark/) (McShane *et al*, 2005). The experiments conformed to the principles set out in the WMA Declaration of Helsinki and the Department of Health and Human Services Belmont Report.

#### Patients
A two-phase design was used to identify the most promising biomarkers associated with PDAC; as a secondary objective, the statistical association between the most promising individual biomarkers and pancreatitis was established. In the first exploratory phase, biomarkers from a small sample of PDAC patients and healthy individuals were screened. In the second confirmatory phase, screened biomarkers were evaluated from a larger indepen-dent sample of PDAC patients, pancreatitis patients, and healthy individuals. Patients with PDAC were matched by sex at a 1:1 ratio to healthy individuals. The statistical procedures applied to each phase are explained as follows:

#### First exploratory phase
Variable clustering was used to identify subsets of biomarkers show-ing an association with PDAC. It was applied in order to control the type I and II error rates with regard to univariate analysis of individ-ual biomarkers.

## The paper explained

### Problem

Reliable biomarkers for the detection of pancreatic ductal adenocarcinoma (PDAC) are still lacking. Their identification is an urgent clinical need to help the management of patients with this aggressive disease. The desmoplastic stromal reaction and extracellular matrix remodeling, typical of pancreatic cancer, can be exploited to identify new diagnostic or prognostic markers for PDAC.

### Results

Seven circulating stroma-related biomarkers were identified associated with PDAC in two independent cohorts of patients, particularly powerful in combination with CA19.9. The association of five of them with early disease was validated in two genetically engineered mouse models. Their relevance as markers of drug response was evidenced in orthotopic patient-derived PDAC xenografts.

### Impact

The identified panels of stroma-related soluble biomarkers are promising for PDAC diagnosis and to monitor treatment response of PDAC patients.

Biomarkers were processed in four steps:

1.  Screen out the candidate biomarkers with more than 10% of value missing (outBks).
2.  Apply a principal components-based clustering method (Black & Watanabe, 2011) to the remaining variables with a 60% explained locus variance threshold.
3.  Test cluster scores for PDAC association in a simple logistic regression framework.
4.  Test singular outBks for PDAC association in a simple logistic regression framework.

In case of complete or almost-complete separation in the univariable logistic regression models, the Wilcoxon rank-sum test to test cluster scores or outBks for PDAC association was used. A $P$-value < 0.10 was taken as indicating a significant association between clusters or outBks and PDAC; only clusters or outBks reaching statistical significance were evaluated in the confirmatory phase.

*Second confirmatory phase*

Screened biomarkers were tested for their association with PDAC in univariate analysis using a logistic regression model stratified by sex; a $P$-value < 0.01 was taken as confirming a significant association between an individual biomarker and PDAC. A multivariable logistic regression model adjusted for age effect was evaluated in order to confirm the significant association (i.e., $P$-value < 0.01) between each selected biomarker and PDAC. As a secondary analysis, the statistical association between biomarkers and pancreatitis was investigated. ROC analysis was used to test the power of screened biomarkers to discriminate between PDAC, pancreatitis, and controls. A multivariable logistic regression model was also developed to improve the discovery power of CA19.9. A forward selection procedure including CA19.9 in every model was used.

A prospective power analysis was performed to determine the sample size of the confirmatory phase. Using a control-to-case ratio of 1:1, 248 patients were necessary to detect relative odds per standard deviation unit equal to 1.61 with an overall two-sided alpha level of 0.01 and power of 0.80. The Shieh–O'Brien approximation was used for power calculations. Assuming an attrition rate of 5%, 262 cases were enrolled.

Statistical analysis was done using SAS version 9.2 (SAS Institute, Cary, NC, USA); the SAS programming code used for the orthoblique principal components-based clustering approach was reported by Black and Watanabe (2011).

### Mouse models

The statistical significance of the differences in plasma biomarker levels was assessed with one-way ANOVA (GEM models) or two-way ANOVA (PDAC-PDX models) followed by Tukey's multiple comparison test. The correlation of biomarker levels with tumor volume was assessed using the Pearson correlation coefficient test. A $P$-value < 0.05 was considered significant. Statistical analysis was performed using GraphPad Prism version 7 software (GraphPad, La Jolla, CA).

**Expanded View** for this article is available online.

### Acknowledgements

We thank Edoardo Micotti for MRI analyses, Roberta Curto for KC and KPC PCR screening and plasma collection, and Judit Baggot and Kerstin Mierke for editing the manuscript. This study was supported by the Italian Association for Cancer Research (AIRC 5X1000 no. 12182 and IG 18853 to RG and IG 16833 to GT), University of Turin-Progetti Ateneo 2014-Compagnia di San Paolo (PANTHER to P.C. and PC-METAIMMUNOTHER to F.N.), and Fondazione Ricerca Molinette. AR is the recipient of a fellowship from AIRC and was supported by Fondazione E. Morandi ONLUS, per lo studio e la cura dei tumori del pancreas.

### Author contributions

AR conducted biomarker analysis in blood samples and performed GEM studies and chronic pancreatitis induction in mice; MRB and AA performed PDAC-PDX studies; LP and VT designed clinical studies and performed statistical analysis; LM performed histological and immunohistochemical analysis; PA, AS, and AF provided human plasma samples; PC and FN provided KC and KPC mice; EM provided human tumor speciments for PDAC-PDX; AR, MB and GT interpreted the data and revised the manuscript; and DB and RG designed the study, supervised the experiments and wrote the manuscript.

### Conflict of interest

The authors declare that they have no conflict of interest.

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
