## [Review Process File · EMBO Molecular Medicine]

Soluble stroma-related biomarkers of pancreatic cancer

Andrea Resovi, Mariarosa Bani, Luca Porcu, Alessia Anastasia, Lucia Minoli, Paola Allavena, Paola Cappello, Francesco Novelli, Aldo Scarpa, Eugenio Morandi, Anna Falanga, Valter Torri, Giulia Taraboletti, Dorina Belotti, Raffaella Giavazzi

Review timeline:	Submission date:	1 December 2017
	Editorial Decision:	29 January 2018
	Revision received:	20 April 2018
	Editorial Decision:	14 May 2018
	Revision received:	25 May 2018
	Accepted:	28 May 2018

Editor: Céline Carret

Transaction Report:

1st Editorial Decision

29 January 2018

Thank you for the submission of your manuscript to EMBO Molecular Medicine. We have now heard back from the three referees whom we asked to evaluate your manuscript.

As you will see from the comments below, the three referees find the study of interest and potentially important for PDAC early detection. However, all three have concerns, sometimes overlapping, that should be addressed in a major revision of this work. We would like you to focus on providing a more detailed experimental characterization of the animals used, including histological analyses, and improve the discussion (on PDAC specificity, of MMP7 role in cancers, of the need for early markers for PDAC for ex.). Upon cross-commenting, it became important to thoroughly show and discuss the early detection aspect, and a proof of concept would be most welcome, otherwise you may have to tune down the title and some of the claims.

We would welcome the submission of a revised version within three months for further consideration and would like to encourage you to address all the criticisms raised as suggested to improve conclusiveness and clarity. Please note that EMBO Molecular Medicine strongly supports a single round of revision and that, as acceptance or rejection of the manuscript will depend on another round of review, your responses should be as complete as possible.

I look forward to receiving your revised manuscript.

***** Reviewer's comments *****

Referee #1 (Remarks for Author):

The work by Resovi et al is aimed at defining new biomarkers for detecting and assessing treatment efficacy in pancreatic cancer. The study focuses on stromal biomarkers in blood plasma and demonstrates a convincing separation between pancreatic cancer and healthy subjects by using a panel of two biomarkers. The biomarkers were confirmed in two different animal models demonstrating elevated levels of three biomarker proteins that correlate with tumor burden. The

work indicates the discovery of potentially new biomarkers that can fulfill more robust detection of pancreatic cancer, but the manuscript is currently associated with several limitations that should be addressed.

Major comments

1) As a starting point for the study, the authors selected 38 candidate biomarkers from previously published work. These candidate biomarkers were extracellular matrix modifying proteins that were detectable in the blood plasma in a first exploratory phase of 41 patients. Many of the selected proteins are however ubiquitously produced in many organs. In fact, several of the proteins are produced in substantially higher amounts in other organs than the pancreas according to the Human Protein Atlas (<https://www.proteinatlas.org/>) and the ProteomicsDB (<https://www.proteomicsdb.org>). This raises the question if these biomarkers are specific to pancreatic cancer or if they are associated with other types of cancers or other diseases that involve inflammation or fibrosis. For example, previous reports have shown that PLG correlates with poor prognosis in renal cancer, MMP7 with poor prognosis in liver and lung cancer and TIMP1 with poor prognosis in renal and colorectal cancer according to the Human Protein Atlas. This concern is supported by the elevated biomarker levels observed in chronic pancreatitis, where the top discriminatory proteins between healthy and pancreatic cancer only show limited discriminatory power between chronic pancreatitis and pancreatic cancer. Low specificity for pancreatic cancer is of considerable importance, as an unspecific response would drastically limit the scope of these markers for detection of pancreatic cancer. This problem is not addressed at any part of the paper nor properly discussed in the discussion section, which substantially weakens the paper.

2) The authors state that there is a profound stromal re-organization during tumor progression, but they provide limited evidence supporting this claim. The study includes animal models to confirm the selected biomarkers during PDAC. In the current version of the manuscript however, the animal models are not characterized in sufficient detail. For instance, the study cites several papers highlighting the importance of for example MMP7 as an inducer of acinar-to-ductal-metaplasia (Crawford et al, 2002; Sawey et al, 2007) and to be focally upregulated in PanIN lesions of the Kras driven-GEM model of PDAC (Hingorani et al, 2003). In this context, it is surprising that the authors have made little efforts to characterize the levels of the most promising biomarkers in the tumors from the animal models. It can be expected that the elevated levels of for instance MMP7 in plasma would follow an increase of MMP7 abundance in the tumors. It is of importance that the authors include quantitative data regarding the abundance and tissue distribution of the biomarkers in the tumors from the animal models. Of especial interest is the correlation between the plasma biomarker abundance levels and the tumor-associated biomarker abundance levels. Furthermore, more detailed information regarding the histology analysis of the pancreas should be added to the manuscript, such as zoomed-out information of the tissue sections and staining's with the selected biomarker candidates.

3) The authors argue that the biomarker panel can be used for early disease detection. Based on the evidence provided in the manuscript these claims are highly questionable. In the animal models, the plasma samples were selected at 150 and 330 days for the two models respectively. However, the survival plots from the original publication by Hingorani et al 2005 seem to show that at least 50% of the animals have succumbed to the disease at the selected time points. Elevated levels of the biomarker panels at this stage of the disease is interesting but it seems unlikely that they can be used for early detection considering the background levels of these biomarkers in healthy plasma. Furthermore, it is surprising that the authors do not display plasma levels of the AUC values for the different stages (1A-IV) that were included in cohort 2 shown in Table 1 and Figure 2. These plots should be added to the manuscript, as a convincing separation between the stages would potentially support the use of these biomarkers for early detection. In addition it would be helpful if the selected time points were motivated in the result section and discussed in more detailed in the discussion section. Lastly, the authors need to provide more compelling evidence that these biomarkers support early detection of pancreatic cancer.

Referee #2 (Remarks for Author):

This is a study to validate stroma related circulating biomarkers in pancreatic cancer by a reputable

team of investigators. They started out with a set of 38 candidates which was filtered down to 12 candidates after an initial triage. Of the latter, seven markers passed the second round of validation in a set of 131 PDAC patients, 30 chronic pancreatitis and 131 healthy subjects that consisted of MMP7, CCN2, IGFBP2, TSP2, sICAM1, TIMP1 and PLG. They determined by means of logistic regression performance in PDAC vs healthy subjects, MMP7+CA19.9, AUC=0.99, 99% CI=0.98-1.00), CCN2+CA19.9, AUC=0.96, 99% CI=0.92-0.99) and for PDAC vs chronic pancreatitis CCN2+PLG+FN+Col4+CA19.9, AUC=0.94, 99% CI=0.88-0.99).

The study in that respect is informative regarding performance in the comparison groups. However it is not clear what the intended clinical application is. The data using mouse models indicated that 5 molecules were associated with PanIN development which they suggest might indicate their utility for early detection. The authors indicate that these markers were also elevated in patient-derived orthotopic PDAC xenografts and associated with response to chemotherapy.

It is not clear how the data using mouse and the data using clinical samples relate to each other in terms of relationship to tumor development, stage and other subject related variables, which leaves it to be determined as to the clinical relevance and applicability of the findings. It is also not clear how the performance in various setting compares with other types of markers and the potential for added contributions of different types of markers.

Referee #3 (Comments on Novelty/Model System for Author):

The authors present a very interesting report on the discovery of a panel of new biomarkers for pancreatic ductal adenocarcinoma. This is a very important discovery. The work represent a potential to possibly predict PDAC at early stage, when patients might be asymptomatic. Clinically this is very important. The authors seem to have validated their findings in KPC mice and compared them with the standard CA19.9 - which is currently inadequate. This review is not quite sure about the emphasis of stromal-derived signature and this would be nice to see clarified. It would also be nice to see the statistical significance of the results given in all the figures themselves (eg Fig 1).

The work is clearly written and very easy to read. In fact this reviewer is pleasantly reassured about the clarity of the work and finds it hard to pick holes in. The work has a simple message, but this is very important. I look forward to the clinical application of the study.

Referee #3 (Remarks for Author):

I think that this is clearly written and important paper. It is simple, so not the sort of thing I usually read for EMBOMolMed, but I like it and it has the potential to be really important.

1st Revision - authors' response

20 April 2018

Referee #1

1)...This raises the question if these biomarkers are specific to pancreatic cancer or if they are associated with other types of cancers or other diseases that involve inflammation or fibrosis. ... This concern is supported by the elevated biomarker levels observed in chronic pancreatitis, where the top discriminatory proteins between healthy and pancreatic cancer only show limited discriminatory power between chronic pancreatitis and pancreatic cancer. This problem is not addressed at any part of the paper nor properly discussed in the discussion section.

We agree with the referee that some of the biomarkers proposed in our study are not exclusively associated with PDAC. In particular, both the two top discriminatory biomarkers MMP7 and CCN2 have been associated with poor prognosis in other types of cancer (Sun Y, Int J Nanomedicine. 2017; Klupp F, BMC Cancer. 2016) and also with fibrotic or inflammatory diseases (Irvine KM, PLoS One. 2016; Dendooven A, Biomark. Biochem. Indic. Expo. Response Susceptibility Chem. 2011). However, our results suggest that improvement in MMP7 and CCN2 specificity result from inclusion of additional biomarkers. The point of our study is that panels of molecules have greater potential as diagnostic biomarkers for PDAC than any single molecule, particularly when evaluated

by PDAC stage. Specifically, MMP7 emerged as an excellent biomarker to discriminate PDAC vs healthy subjects in combination with CA19.9 with an AUC of 0.99 (99% CI=0.98-1.00) and CCN2 complemented well the performance of CA19.9 in distinguishing PDAC from healthy subjects. Further addition of PLG, FN and Col4 to the CCN2 and CA19.9 panel increased the performance of CA19.9 in distinguishing PDAC from CP.

These data are described at page 8, presented in Figure 2B Figure EV1 and Table EV4, and discussed between page 12 and 13.

2) The study includes animal models to confirm the selected biomarkers during PDAC. ... It is of importance that the authors include quantitative data regarding the abundance and tissue distribution of the biomarkers in the tumors from the animal models. ... Of especial interest is the correlation between the plasma biomarker abundance levels and the tumor-associated biomarker abundance levels. ... Furthermore, more detailed information regarding the histology analysis of the pancreas should be added to the manuscript, such as zoomed-out information of the tissue sections and staining's with the selected biomarker candidates.

Following the reviewer indication, we have added new data showing the parallel analysis of tumor expression and plasma levels of three of the best candidates, TIMP1, MMP7 and TSP2, in the GEM model and PDAC-PDX.

KC-GEM model:

TIMP1, MMP7 and TSP2 tissue expression (measured by real-time PCR) paralleled plasma levels (measured by ELISA) and were significantly higher in KC than control PdxCre mice. Both soluble and tissue markers were upregulated in association with development of PanIN-1A, PanIN-1B and PanIN-2 lesions, evaluated by histological analysis of the pancreas in these mice (Figure 3B and Appendix Figure S2B).

Immunohistological analysis was performed to analyze the distribution of the biomarkers in the tumor tissue. TIMP1 showed a progressive increase of stromal staining along with a moderate expression by pancreatic ductal cells in PanIN lesions. MMP7 was moderately to markedly expressed by ductal epithelial cells, and the intensity of staining increased along with the progression of the disease. Diffuse, mild to moderate expression of TSP2 was observed within the stroma surrounding PanIN lesions.

In addition, real-time PCR confirmed that also the expression of CCN2 and ICAM1, though not IGFBP2 and PLG, were increased in GEM PanIN lesions. Progressive increase of CCN2 staining in PanIN lesions was also confirmed by immunohistochemical analysis.

These results are shown on figure 3B, C and Appendix Figure S2B and described on page 8 and 9.

PDAC-PDX:

Real-time PCR analysis confirmed the increased expression of murine (stroma-derived) TIMP1, TSP2, CCN2, ICAM1 in HuPa4, HuPa8 and HuPa11 tumor tissues compared with healthy mice. The increase in tissue TIMP1 and TSP2 paralleled their plasma level, significantly higher in all three PDAC-PDX compared to healthy mice. Notably, no increase in the expression of MMP7 was detected by RT-PCR, suggesting that the increased levels of MMP7 detected in the plasma of tumor-bearing mice might reflect a systemic response of the host to the tumor, in line with the known association of MMP7 with inflammation.

Immunohistological analysis confirmed that, in agreement with the real-time PCR results, moderate to marked increase in TIMP1, TSP2, CCN2 were found associated with the stroma of PDAC-PDX. These results are shown on figure 4A, C, D, described on pages 10 and discussed on pages 13.

3) The authors need to provide more compelling evidence that these biomarkers support early detection of pancreatic cancer. ... In the animal models, the plasma samples were selected at 150 and 330 days for the two models respectively. However, the survival plots from the original publication by Hingorani et al 2005 seem to show that at least 50% of the animals have succumbed to the disease at the selected time points. ... It would be helpful if the selected time points were motivated in the result section and discussed in more detailed in the discussion section.

As requested by the reviewer, we provide data on the analysis of TIMP1, MMP7 and TSP2 at earlier time points in the GEM models: plasma samples were analyzed at 60, 120, 180, 240 and 330 days of age in KC mice and at 30, 90 and 150 days of age in KPC mice (Figure 3A). The levels of the three biomarkers increased with time, concurrently with PanIN development suggesting that they appear in the early pre-invasive stage of the disease and further supporting the value of these markers for early detection of pancreatic cancer.

Results are described at the end of page 8 and on page 9 and discussed on page 13.

Concerning the differences in tumor development and survival between our GEM models and those described in the original publication by Hingorani, it should be noted that the two GEM strains used in this study were ri-derived from the original ones by crossing single-mutated Kras^{G12D} or double-mutated Kras^{G12D} and Trp53^{R172H} with C57BL/6 mice expressing Cre Recombinase. These mice develop pancreatic carcinoma with different kinetics (Cappello P, Gastroenterology. 2013). At day 330, 100% of KC mice were still alive and displayed pancreatic early lesions with difference in PanIN grade. Histological analysis is displayed in Appendix Figure S2A.

Furthermore, it is surprising that the authors do not display plasma levels of the AUC values for the different stages (IA-IV) that were included in cohort 2 shown in Table 1 and Figure 2. These plots should be added to the manuscript, as a convincing separation between the stages would potentially support the use of these biomarkers for early detection.

We thank the reviewer for suggesting this analysis to support the value of the biomarker panels for early detection. In the second cohort of patients, we found that TIMP1, sicam1, MMP7, TSP2, PLG, IGFBP2 and CCN2 were indeed significantly up regulated at early stages (stage IA, IB and IIA) when tumor is confined to the pancreas and is not spread to nearby lymph nodes (N0) or to distant sites (M0). AUCs used to estimate the predictive accuracy of distributional models didn't change significantly between stages, supporting the potential use of these biomarkers for early detection. Data are described on page 7, AUC values for the different stages are shown in Figure EV1 and Appendix Table S2.

Multivariable models defined in the overall population confirmed their optimal predictive accuracy without significant interaction between stages. This is described on page 8 and Appendix Table S2.

Referee #2

...it is not clear what the intended clinical application is. The data using mouse models indicated that 5 molecules were associated with PanIN development which they suggest might indicate their utility for early detection. The authors indicate that these markers were also elevated in patient-derived orthotopic PDAC xenografts and associated with response to chemotherapy. It is not clear how the data using mouse and the data using clinical samples relate to each other in terms of relationship to tumor development, stage and other subject related variables, which leaves it to be determined as to the clinical relevance and applicability of the findings. It is also not clear how the performance in various setting compares with other types of markers.

We have modified the manuscript, added new data, and improved the integration between data from patients and mouse models to clarify the relevance and possible clinical application of the identified biomarker panels.

Concerning the intended clinical application, our study suggests that the identified stroma-related molecules, associated with PDAC, are exploitable as diagnostic biomarkers and as end-points of targeted therapies.

As underlined in the response to referee #1 (points 1 and 3) the expression of biomarkers has been analyzed in patients at early stage of the disease (second cohort). The finding that TIMP1, sicam1, MMP7, TSP2, PLG, IGFBP2 and CCN2 were significantly up regulated at stages IA, IB and IIA supports the value of these biomarkers for early detection. We show that AUCs used to estimate the predictive accuracy of the distributional models did not change significantly between the stages of the disease, further supporting the use of these biomarkers for early detection. This is described on page 8, Figure EV1 and Appendix Table S2. Specifically, MMP7 emerged as an excellent biomarker to discriminate PDAC vs healthy subjects, particularly in combination with CA19.9 and CCN2 complemented well the performance of CA19.9 in distinguishing PDAC from healthy subjects. Further addition of PLG, FN and Col4 to the CCN2 and CA19.9 panel increased the performance of CA19.9 in distinguishing PDAC from CP. These data are discussed on pages 12-13. It was also verified that age, another relevant subject-related variable, did not affect the significant association between each selected biomarker and PDAC, as assessed by a multivariable logistic regression model, adjusted for age effect. Data are described on page 7.

Concerning the integration between the findings in mouse and clinical samples, the above data on early stage clinical samples are in complete agreement with findings obtained in the GEM models of progressive PDAC development, that showed significantly higher TIMP1, MMP7 and TSP2 plasma levels in association with early tumor progression.

Studies with PDAC-PDX further indicate that TIMP1, MMP7, and TSP2 correlated with tumor burden and drug response. This suggests they might be also useful biomarkers to assess treatment efficacy. As mentioned on page 15, this hypothesis must be further validated in the clinical setting.

We have compared the performance of our biomarkers with CA19.9, the reference for PDAC. As described on page 7, the differences between PDAC and healthy subjects were excellent for MMP7 (AUC=0.98) and good for CCN2 (AUC=0.86), IGFBP2 and TIMP1 (AUC=0.82), which demonstrated a discriminatory ability similar to CA19.9 (AUC=0.87) while TSP2 (AUC=0.78), sICAM1 (AUC=0.77) and PLG (AUC=0.66) had a weaker discriminatory ability (Figure EV1 and Table EV4). AUCs used to estimate the predictive accuracy of distributional models didn't change significantly between stages (Figure EV1 and Appendix Table S2). As discussed on page 15, prospective confirmatory clinical studies are required to validate them as diagnostic PDAC biomarkers complementing CA19.9.

Referee #3

The authors present a very interesting report on the discovery of a panel of new biomarkers for pancreatic ductal adenocarcinoma. This is a very important discovery. The work represents a potential to possibly predict PDAC at early stage, when patients might be asymptomatic. Clinically this is very important. The authors seem to have validated their findings in KPC mice and compared them with the standard CA19.9 - which is currently inadequate.

We thank the referee for the positive comments.

This reviewer is not quite sure about the emphasis of stromal-derived signature and this would be nice to see clarified.

The hypothesis at the basis of this study was that molecules associated with the unique stroma desmoplastic reaction typically associated with PDAC progression might represent sensible and valuable biomarkers of this disease. The new data added to the revised manuscript (see also response to referee #1) further support this concept, proving that these markers are upregulated also in early stages of the disease in patients. Moreover, the new analysis of marker expression (real-time PCR and IHC) in GEM and PDAC-PDX confirms that, with the exception of MMP7, these markers are indeed associated with the tumor stroma. These new findings are described on pages 9 and 10, Appendix Figure S2B, Figure 3C, 4C, 4D, and their meaning addressed in the discussion.

It would also be nice to see the statistical significance of the results given in all the figures themselves (eg Fig 1).

Statistical significance values have been added to all the figures.

2nd Editorial Decision

14 May 2018

Thank you for the submission of your revised manuscript to EMBO Molecular Medicine. We have now received the enclosed reports from the referees that were asked to re-assess it. As you will see the reviewers are now supportive and I am pleased to inform you that we will be able to accept your manuscript pending a few final editorial amendments.

***** Reviewer's comments *****

Referee #1 (Remarks for Author):

The authors have addressed all concerns.

Referee #2 (Remarks for Author):

In this revised manuscript the authors have addressed some of the issues brought up by reviewers. Although the clinical application(s) that may result from this work are debatable, the manuscript is likely to be of interest to investigators in the field

Corresponding Author Name: Raffaella Giavazzi and Dorina Belotti

Manuscript Number: EMM-2017-08741